# POLICY-INDUCED SELF-SUPERVISION IMPROVES REPRESENTATION FINETUNING IN VISUAL RL

## ABSTRACT

We study how to transfer representations pretrained on source tasks to target tasks in visual percept based RL. We analyze two popular approaches: freezing or finetuning the pretrained representations. Empirical studies on a set of popular tasks reveal several properties of pretrained representations. First, finetuning is required even when pretrained representations perfectly capture the information required to solve the target task. Second, finetuned representations improve learnability and are more robust to noise. Third, pretrained bottom layers are task-agnostic and readily transferable to new tasks, while top layers encode task-specific information and require adaptation. Building on these insights, we propose a self-supervised objective that *clusters representations according to the policy they induce*, as opposed to traditional representation similarity measures which are policy-agnostic (*e.g.* Euclidean norm, cosine similarity). Together with freezing the bottom layers, this objective results in significantly better representation than frozen, finetuned, and self-supervised alternatives on a wide range of benchmarks.

## 1 INTRODUCTION

Learning representations via pretraining is a staple of modern transfer learning. Typically, a feature encoder is pretrained on one or a few source task(s). Then it is either frozen (*i.e.*, the encoder stays fixed) or finetuned (*i.e.*, the parameters of the encoders are to be updated) when solving a new downstream task (Yosinski et al., 2014). While whether to freeze or finetune is application-specific, finetuning outperforms in general freezing when there are sufficient (labeled) data and compute. This pretrain-then-transfer recipe has led to many success stories in vision (Razavian et al., 2014; Chen et al., 2021), speech (Amodei et al., 2016; Akbari et al., 2021), and NLP (Brown et al., 2020; Chowdhery et al., 2022).

For reinforcement learning (RL), however, finetuning is a costly option as the learning agent needs to collect its own data specific to the downstream task. Moreover, when the source tasks are very different from the downstream task, the first few updates from finetuning destroy the representations learned on the source tasks, cancelling all potential benefits of transferring from pretraining. For those reasons, practitioners often choose to freeze representations, thus completely preventing finetuning.

But representation freezing has its own shortcomings, especially pronounced in visual RL, where the (visual) feature encoder can be pretrained on existing image datasets such as ImageNet (Deng et al., 2009) and even collections of web images. Such generic but easier-to-annotate datasets are not constructed with downstream (control) tasks in mind and the pretraining does not necesarily capture important attributes used to solve those tasks. For example, the downstream embodied AI task of navigating around household items (Savva et al., 2019; Kolve et al., 2017) requires knowing the precise size of the objects in the scene. Yet this information is not required when pretraining on visual objection categorization tasks, resulting in what is called *negative transfer* where a frozen representation hurts downstream performance.

More seriously, even when the (visual) representation needed for the downstream task is known *a priori*, it is unclear that learning it from the source tasks then freezing it should be preferred to finetuning, as shown in Figure 1. On the left two plots, freezing representations (`Frozen`) underperforms learning representations using only downstream data (`De Novo`). On the right two plots, we observe the opposite outcome. Finetuning representations (`Finetuned`) performs well overall, but fails to unequivocally outperform freezing on the rightmost plots.

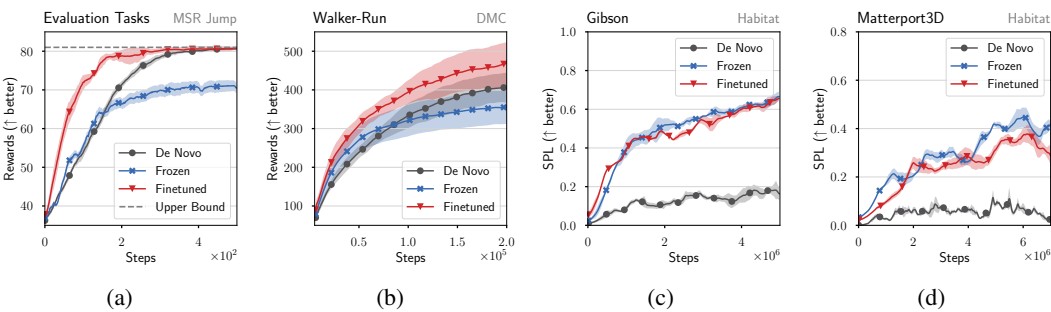

Figure 1: **When should we freeze or finetune pretrained representations in visual RL?** Reward and success weighted by path length (SPL) transfer curves on `MSR Jump`, `DeepMind Control`, and `Habitat` tasks. Freezing pretrained representations can underperform no pretraining at all (Figures 1a and 1b) or outperform it (Figures 1c and 1d). Finetuning representations is always competitive, but fails to significantly outperform freezing on visually complex domains (Figures 1c and 1d). Solid lines indicate mean over 5 random seeds, shades denote 95% confidence interval. See Section 3.2 for details.

**Contributions**   *When should we freeze representations, when do they require finetuning, and why?* This paper answers those questions through several empirical studies on visual RL tasks ranging from simple game and robotic tasks (Tachet des Combes et al., 2018; Tassa et al., 2018) to photo-realistic `Habitat` domains (Savva et al., 2019).

Our studies highlight properties of finetuned representations which improve learnability. First, they are more consistent in clustering states according to the actions they induce on the downstream task; second, they are more robust to noisy state observations. Inspired by these empirical findings, we propose `PiSCO`, **a representation finetuning objective which encourages robustness and consistency with respect to** *the actions they induce*.

We also show that visual percept feature encoders first compute task-agnostic information and then refine this information to be task-specific (*i.e.*, predictive of rewards and / or dynamics) — a well-known lesson from computer vision (Yosinski et al., 2014) but, to the best of our knowledge, never demonstrated for RL so far. We suspect that finetuning with RL destroys the task-agnostic (and readily transferrable) information found in lower layers of the feature encoder, thus cancelling the benefits of transfer. To retain this information, we show how to identify transferrable layers, and propose to **freeze those layers while adapting the remaining ones** with `PiSCO`. This combination yields excellent results on all testbeds, outperforming both representation freezing and finetuning.

## 2   RELATED WORKS AND BACKGROUND

Learning representations for visual reinforcement learning (RL) has come a long way since its early days. Testbeds have evolved from simple video games (Koutník et al., 2013; Mnih et al., 2015) to self-driving simulators (Shah et al., 2018; Dosovitskiy et al., 2017), realistic robotics engines (Tassa et al., 2018; Makoviychuk et al., 2021), and embodied AI platforms (Savva et al., 2019; Kolve et al., 2017). Visual RL algorithms have similarly progressed, and can now match human efficiency and performance on the simpler video games (Hafner et al., 2021; Ye et al., 2021), and control complex simulated robots in a handful of hours (Yarats et al., 2022; Laskin et al., 2020; Hansen et al., 2022).

In spite of this success, visual representations remain challenging to transfer in RL (Lazaric, 2012). Prior work shows that learned representations can be surprisingly brittle, and fail to generalize to minor changes in pixel observations (Witty et al., 2021). This perspective is often studied under the umbrella of generalization (Zhang et al., 2018; Cobbe et al., 2020; Packer et al., 2018) or adversarial RL (Pinto et al., 2017; Khalifa et al., 2020). In part, those issues arise due to our lack of understanding in what can be transferred and how to learn it in RL. Others have argued for a plethora of representation pretraining objectives, ranging from capturing policy values (Lehnert et al., 2020; Liu et al., 2021) and summarizing states (Mazoure et al., 2020; Schwarzer et al., 2021; Abel et al., 2016; Littman & Sutton, 2001) to disentangled (Higgins et al., 2017) and bi-simulation metrics (Zhang et al., 2021; Castro & Precup, 2010). Those objectives can also be aggregated in the hope of learning more generic representations (Gelada et al., 2019; Yang & Nachum, 2021). Nonetheless, it remains unclear which of these methods should be preferred to learn generic transferrable representations.

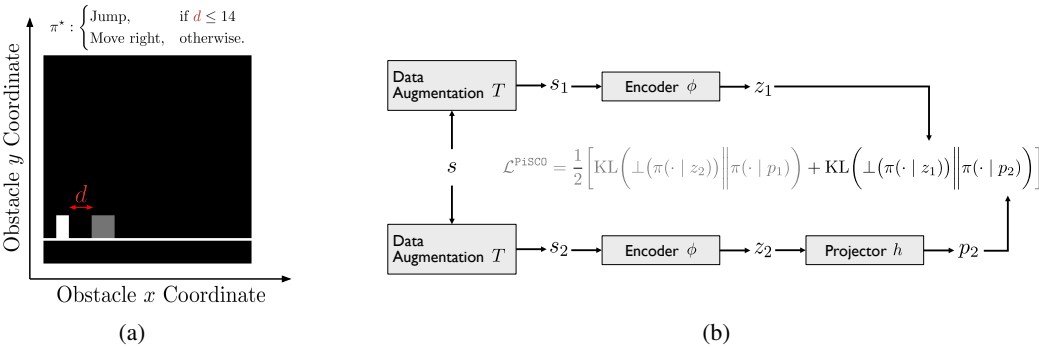

(a)                                                          (b)

Figure 2: **(a) Annotated observation from `MSR Jump`.** On this simple game, the optimal policy consists of moving right until the agent (white) is 14 pixels away from the obstacle (gray), at which point it should jump. **(b) Diagram for our proposed `PiSCO` consistency objective.** It encourages the policy's actions to be consistent for perturbation of state $s$, as measured through the actions induced by the policy. See Section 4 for details.

Those issues are exacerbated when we pretrain representations on generic vision datasets (*e.g.*, ImageNet (Deng et al., 2009)), as in our `Habitat` experiments. Most relevant to this setting are the recent works of Xiao et al. (2022) and Parisi et al. (2022): both pretrain their feature encoder on real-world images and freeze it before transfer to robotics or embodied AI tasks. However, neither reports satisfying results as they both struggle to outperform a non-visual baseline built on top of proprioceptive states. This observation indicates that more work is needed to successfully apply the pretrain-then-transfer pipeline to visual RL. A potentially simple source of gain comes finetuning those generic representations to the downstream task, as suggested by Yamada et al. (2022). While this is computationally expensive, prior work has hinted at the utility of finetuning in visual RL (Schwarzer et al., 2021; Wang et al., 2022).

In this work, we zero-in on representation freezing and finetuning in visual RL. Unlike prior representation learning work, our goal is not to learn generic representations that transfer; rather, we would like to specialize those representations to a specific downstream task. We contribute in Section 3 an extensive empirical analysis highlighting important properties of finetuning, and take the view that finetuning should emphasize features that easily discriminate between good and bad action on the downstream task — in other words, *finetuning should ease decision making*. Building on this analysis, we propose a novel method to simplify finetuning, and demonstrate its effectiveness on challenging visual RL tasks where naive finetuning fails to improve upon representation freezing.

## 3  UNDERSTANDING WHEN TO FREEZE AND WHEN TO FINETUNE

We presents empricial analysis of representation freezing and finetuning. In Sections 3.1 and 3.2 we detail the results from Figure 1 mentioned above. Then, we show in Section 3.3 a surprising result on the `MSR Jump` tasks: although freezing the pretrained representations results in negative transfer, those representations are sufficiently informative to perfectly solve the downstream task. This result indicates that capturing the relevant visual information through pretraining is not always effective for RL transfer. Instead, Section 3.4 shows that representations need to emphasize task-relevant information to improve learnability – an insight we explore in Section 3.5 and build upon in Section 4 to motivate a simple and effective approach to finetuning.

### 3.1  EXPERIMENTAL TESTBEDS

Our testbeds are standard and thoroughly studied RL tasks in literature. Our description here focuses on using to study RL transfer. We defer details and implementations to Appendix B.

**MSR Jump** (Tachet des Combes et al., 2018) The agent (a gray block) needs to cross the screen from left to right by jumping over an obstacle (a white block). The agent's observations are video game frames displaying the entire world state (see Figure 2a) based on which it can choose to "move right" or "jump". We generate 140 tasks by changing the position of the obstacle vertically and horizontally; half of them are used for pretraining, half for evaluation. We train the convolutional actor-critic agent from Mnih et al. (2016) with PPO (Schulman et al., 2017) for 500 iterations on all pretraining tasks, and transfer its 4-layer feature encoder to jointly solve all evaluation tasks. Like pretraining, we train the actor and critic heads of the agent with PPO for 500 iterations during transfer.

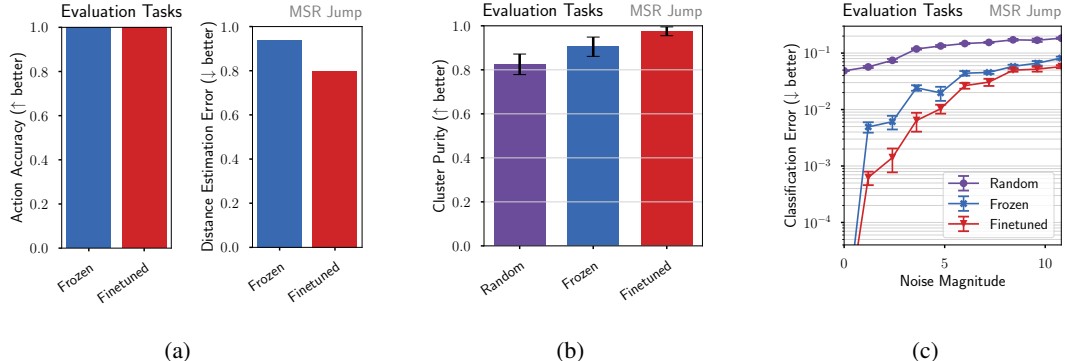

(a)             (b)             (c)

Figure 3: **(a) Transfer can still fail even though frozen representations are informative enough.** On `MSR Jump` tasks, we can perfectly regress the optimal actions (accuracy = 1.0) and agent-obstacle distance (mean square error < 1.0) with *frozen or finetuned* representations. Combined with Figure 1a, those results indicate that *capturing the right information is not enough for RL transfer*. **(b) Finetuned representations yield purer clusters.** Given a state, we measure the expected purity of the cluster consisting of the 5 closest neighbours of that state in representation space. For `Finetuned` representations, this metric is significantly higher (98.75%) than for `Frozen` (91.41%) or `Random` (82.98%), showing that states which beget the same actions are closer to one another and thus easier to learn. **(c) Finetuned representations are more robust to perturbations.** For source and downstream tasks, the classification error (1 - action accuracy) degrades significantly more slowly for `Finetuned` representations than for `Frozen` ones under increasing data augmentation noise, suggesting that robustness to noise improves learnability. See Section 3.3 for details.

**DeepMind Control** (DMC) (Tassa et al., 2018) This robotics suite is based on the MuJoCo (Todorov et al., 2012) physics engine. We use visual observations as described by Yarats et al. (2021), and closely replicate their training setup. The learning agent consists of action-value and policy heads, together with a shared convolutional feature encoder whose representations are used *in lieu of* the image observations. We pretrain with DrQ-v2 (Yarats et al., 2022) on a single task from a given domain, and transfer only the feature encoder to a different task from the same domain. For example, our `Walker` agent is pretrained on `Walker-Walk` and transfers to `Walker-Run`.

**Note:** The main text uses `Walker` to illustrate results on `DeepMind Control`, and more tasks are included in Appendix C (including `Cartpole Balance`→`Swingup` and `Hopper Stand`→`Hop`).

**Habitat** (Savva et al., 2019) In this setting, we pretrain a ConvNeXt (tiny) (Liu et al., 2022) to classify images from the ImageNet (Deng et al., 2009) dataset. We use the "trunk" of the ConvNeXt as a feature encoder (*i.e.*, discard the classification head) and transfer it to either Gibson (Shen et al., 2021) or Matterport3D (Chang et al., 2017) scenes simulated with the `Habitat` embodied AI simulator. The tasks in `Habitat` consist of navigating to a point from an indoor scene (*i.e.*, PointNav) from visual observations, aided with GPS and compass sensing. Our implementation replaces the residual network from "Habitat-Lab" (Szot et al., 2021) with our ConvNeXt[1], and uses the provided DDPPO (Wijmans et al., 2020) implementation for transfer training. We train for 5M steps on Gibson and 7M steps on Matterport3D, include all scenes from the respective datasets, and report reward weighted by path length (SPL).

## 3.2 WHEN FREEZING OR FINETUNING WORKS

As a first experiment, we take the pretrained feature encoders from the previous section and transfer them to their respective downstream tasks. We consider three scenarios.

- `De Novo`: where the feature encoder is randomly initialized (*not* pretrained) and its representations are learned on the downstream task only. The feature encoder might use different hyper-parameters (*e.g.*, learning rate, momentum, etc) from the rest of the agent.
- `Frozen`: where the feature encoder is pretrained and its representations are frozen (*i.e.*, we never update the weights of the feature encoder).
- `Finetuned`: where the feature encoder is pretrained and its representations are finetuned (*i.e.*, we update the weights of the feature encoder using the data and the algorithm of the downstream task). As for `De Novo`, the hyper-parameters for the feature encoder are tuned separately from the other hyper-parameters of the model.

---

[1]We choose a ConvNeXt because it removes batch normalization and dropout layers after pretraining.

Figure 1 displays those results. Comparing `De Novo` and `Frozen` shows that freezing representations works well on `Habitat` (see Figures 1c and 1d), likely because these tasks demand significantly richer visual observations. However, freezing representations completely fails on the simpler `MSR Jump` and `DeepMind Control` tasks (see Figures 1a and 1b). The pretrained feature encoders don't provide any benefit over a randomly initialized ones, and even significantly hurt transfer on `MSR Jump`. The latter is especially surprising because there is very little difference between pretraining and evaluation tasks (only the obstacle position is changed). Those results beg the question: *why does transfer fail so badly on `MSR Jump`?*

On the other hand, `Finetuned` performs reasonably well across all benchmarks but those results hide an important caveat: finetuning can sometimes collapse to representation freezing. This was the case on `Habitat`, where our hyper-parameter search resulted in a tiny learning rate for the feature encoder ($1e^{-6}$, the smallest in our grid-search), effectively preventing the encoder to adapt. We hypothesize that on those large-scale and visually challenging tasks, `Finetuned` offers no improvement over `Frozen` because RL updates are too noisy for finetuning. Thus, we ask: *how can we help stabilize finetuning in RL?*

We answer both questions in the remaining of this paper. We take a closer look at `MSR Jump` tasks and explain why transfer failed. We show that a core role of finetuning is to emphasize task-specific information, and that layers which compute task-agnostic information can be frozen without degrading transfer performance. In turn, this suggests an instinctive strategy to help stabilize RL training: freezing task-agnostic layers, and only finetuning task-specific ones.

## 3.3 FREEZING FAILS EVEN WHEN LEARNED REPRESENTATIONS ARE USEFUL

We now show that, on `MSR Jump`, the pretrained representations are informative enough to perfectly solve all evaluation tasks.

Due to the simplicity of the dynamics, we can manually devise the following optimal policy because the size of the agent, the size of the obstacle, and the jump height are shared across all tasks:

$$\pi^* = \begin{cases} \text{Jump,} & \text{if the agent is 14 pixels to the left of the obstacle.} \\ \text{Move right,} & \text{otherwise.} \end{cases}$$

We empirically confirmed this policy reaches the maximal rewards (81) on all pretraining and evaluation tasks. Immediately, we see that the distance from the agent to the obstacle is the only information that matters to solve `MSR Jump` tasks. This observation suggests the following experiment.

To measure the informativeness of pretrained representations, we regress the distance between the agent and the obstacle from pretrained representations. If the estimation error (*i.e.*, mean square error) is lower than 1.0 pixel, the representations are informative enough to perfectly solve `MSR Jump` tasks. To further validate this scenario, we also regress the optimal actions from the simple policy above given pretrained representations. Again, if the acurracy is perfect the representations are good enough. We repeat this experiment for finetuned representations, and compare the results against one another.

Figure 3a reports results for pretrained and finetuned (*i.e.*, after transfer) representations. We regress the distance and optimal actions with linear models from evaluation task observations, using a mean square error and binary cross-entropy loss. Surprisingly, *both* pretrained and finetuned representations can perfectly regress the distance and optimal actions: they get sub-pixel estimation error (0.94 and 0.79, respectively) and perfect accuracy (both 100%). Combined with the previous section, these results indicate that capturing the right visual information is not enough for successful transfer in RL. They also put forward the following hypothesis: *one role of finetuning is to emphasize information that eases decision making.* This hypothesis is supported by the lower distance error for `Finetuned` in the right panel of Figure 3a.

## 3.4 FINETUNING IMPROVES LEARNABILITY AND ROBUSTNESS TO NOISE

How is information refined for decision making? Our next set of experiments answers this question by highlighting differences between pretrained and finetuned representations.

First, we measure the ability of each feature encoder to cluster states assigned to identical actions together. Intuitively, we expect states that require similar actions to have similar representations.

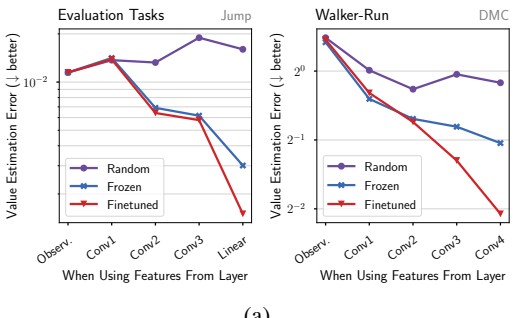
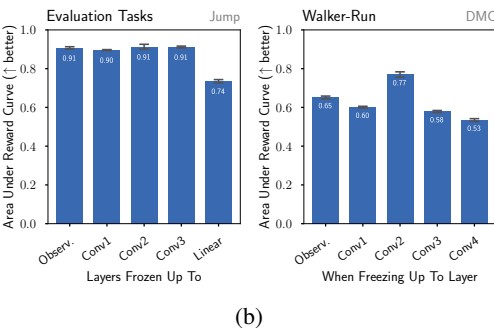

(a)                                               (b)

Figure 4: **Frozen layers that retain the same information as finetuned layers can be kept frozen. (a)** Mean squared error when linearly regressing action values from representations computed by a given layer. Some early layers are equally good at predicting action values before or after finetuning. This suggests those layers can be kept frozen, potentially stabilizing training. See Section 3.5 for details. **(b)** Area under the reward curve when freezing up to a given layer and finetuning the rest. Freezing early layers does not degrade performance, and sometimes improves it (*e.g.*, up to `Conv2` on `DeepMind Control`).

To that end, we compute the average purity when clustering a state with its 5 closest neighbors in representation space. In other words, given state $s$, we find the 5 closest neighbors to $x$ in representation space and count how many of them are assigned the same optimal action as $s$. Figure 3b the average cluster purity for random, pretrained, and finetuned representations. We find that `Finetuned` representations yield clusters with significantly higher purity ($98.75\%$) than `Frozen` ($91.41\%$) or random ($82.98\%$) ones, which explains why PPO converges more quickly when representations are finetuned: it is easier to discriminate between the states requiring "jump" *v.s.* "move right", if the representations for those states cluster homogenously.

Second, we uncover another property of finetuned representations, namely, that they are more robust to perturbations. We replicate the optimal action classification setup from Figure 3a but perturb states randomly rotating them. For each image, the degree of the rotation is randomly sampled from range $(-x, x)$. Figure 3c shows how classification accuracy degrades as we let $x$ increase from $0°$ to $12°$. Both `Frozen` and `Finetuned` degrade with more noise, but `Finetuned` is more robust and degrades more slowly.

Those results point to finetuned representations that consistently map to similar actions and are robust to perturbations They set the stage for Section 4 where we introduce `PiSCO`, a policy consistency objective which builds on these insights. In the next section, we continue our analysis and further investigate how information gets refined *in each layer* of the feature encoder.

## 3.5   WHEN AND WHY IS REPRESENTATION FINETUNING REQUIRED?

As the previous section uncovered important pitfalls of representation freezing, we now turn to finetuning in the hope of understanding why it succeeds when it does. To that end, we dive into the representations of individual layers in `Finetuned` feature encoders and compare them to the `Frozen` and random representations. We take inspiration from the previous section and hypothesize that "purpose drives adaptation"; that is, a layer only needs finetuning if it computes task-specific information that eases decision making on the downstream task. Conversely, layers that compute task-agnostic information can be frozen thus potentially stabilizing RL training. To test this hypothesis, we conduct two layer-by-layer experiments: linear probing and incremental freezing.

With layer-by-layer linear probing, we aim to measure how relevant each layer is when choosing an action (Ding et al., 2021; Zhang et al., 2022). To start, we collect a dataset of $10,000$ state-action pairs and their corresponding action values with the `Finetuned` policy on the downstream task. The action values are computed with an action value function trained to minimize the temporal difference error on the collected state-action pairs (more details in Appendix C). We then proceed as follows for each layer $l$ in the feature encoder. First, we store the representations computed by layer $l$, and extract their most salient features by projecting them to a 50-dimensional vector with PCA. Second, we regress action values from the projected representations and their corresponding actions using a linear regression model. Finally, we report the mean squared error for this linear model at layer $l$. Intuitively, the lower the linear probing error the easier it is to predict how good an action is for a

given state[2]. We repeat this entire process for `Frozen` and `Finetuned` feature encoders, as well as one with a randomly initialized weights (*i.e.*, `Random`).

Results for this experiment are shown in Figure 4a. On all testbeds, both `Frozen` and `Finetuned` curves trend downwards and lie significantly lower than `Random`, indicating, as expected, that representations specialize for decision making as we move up the feature encoder. We also notice that the first few layers of `Frozen` and `Finetuned` track each other closely before `Frozen` starts to stagnate. On `MSR Jump`, this happens after the last convolutional layer (`Conv3`) and on `DeepMind Control` after the second one (`Conv2`). This evidence further supports our previous insight that early layers compute task-agnostic information, and suggests a new hypothesis: *can we freeze the early layers without degrading finetuning performance?* If so, this strategy might help stabilize finetuning in visually complex tasks like `Habitat`.

We confirm this hypothesis by combining freezing and finetuning in our second layer-by-layer experiment. For each target layer $l$, we take the pretrained feature encoder and freeze all layers up to and including $l$; the remaning layers are finetuned using the same setup as Figure 1.

Figure 4b summarizes this experiment with the (normalized) area under the reward curve (AURC). We preferred this metric over "highest reward achieved" since the latter does not consider how quickly the agent learns. On `MSR Jump`, the pretrained layers that have similar value estimation error in Figure 4a can be frozen without degrading adaptation. But, freezing yields lower AURC when value estimation error stagnates (as for `Linear`). Similarly, freezing the last two layers on `DeepMind Control` (`Conv3` and `Conv4`, which did not match `Finetuned`'s value estimation error) also degrades performance. We also see that adapting too many layers (*i.e.*, when `Conv1` is frozen but not `Conv2`) reduces the AURC. The training curves show this is due to slower convergence, suggesting that `Conv2` already computes useful representations which can be readily used for the downstream task.

Merging the observations from our layer-by-layer experiments, we conclude that *pretrained layers which extract task-agnostic information can be frozen*. We show in Section 5 that this conclusion significantly helps stabilize training when finetuning struggles, as in `Habitat`.

## 4 FINETUNING WITH A POLICY-INDUCED SELF-SUPERVISED OBJECTIVE

Section 3.4 suggests that an important outcome of finetuning is that states which are assigned the same actions cluster together. This section builds on this insight and introduces `PiSCO`, a self-supervised objective which attempts to accelerate the discovery of representations which cluster together for similar actions. At a high level `PiSCO` works as follows: given a policy $\pi$, it ensures that $\pi(a \mid s_1)$ and $\pi(a \mid s_2)$ are similar if states $s_1$ and $s_2$ should yield similar actions (*e.g.*, $s_1, s_2$ are perturbed version of state $s$). This objective is self-supervised because it applies to *any* policy $\pi$ (not just the optimal one) and doesn't require knowledge of rewards nor dynamics.

More formally, we assume access to a batch of states $\mathcal{B}$ from which we can sample state $s$. Further, we compute two embedding representations for state $s$. The first, $z = \phi(s)$, is obtained by passing $s$ through an encoder $\phi(\cdot)$; the second, $p = h(z)$, applies a projector $h(\cdot)$ on top of representation $z$. Both represenations $z$ and $p$ have identical dimensionality and can thus be used to condition the policy $\pi$. The core contribution behind `PiSCO` is to measure the dissimilarity between $z$ and $p$ in terms of the distribution they induce through $\pi$:

$$\mathcal{D}(z, p) = \text{KL}\left(\perp\left(\pi(\cdot \mid z)\right) \| \pi(\cdot \mid p)\right),$$

where $\text{KL}(\cdot \| \cdot)$ is the Kullback-Leibler divergence, and $\perp(x)$ is the stop-gradient operator, which sets all partial derivatives with respect to $x$ to 0. The choice of the Kullback-Leibler is arbitrary — other statistical divergence are valid alternatives.

The final ingredient in `PiSCO` is a distribution $T(s' \mid s)$ over perturbed states $s'$. This distribution is typically implemented by randomly applying a set of data augmentation transformations on the state $s$. Then, the `PiSCO` objective (for **P**olicy-**i**nduced **S**elf-**C**onsistency **O**bjective) is to minimize the

---

[2]We found action values to be better targets to measure ease of decision making than, say, action accuracy which is upper bounded by 1.0 as in Figure 3a.

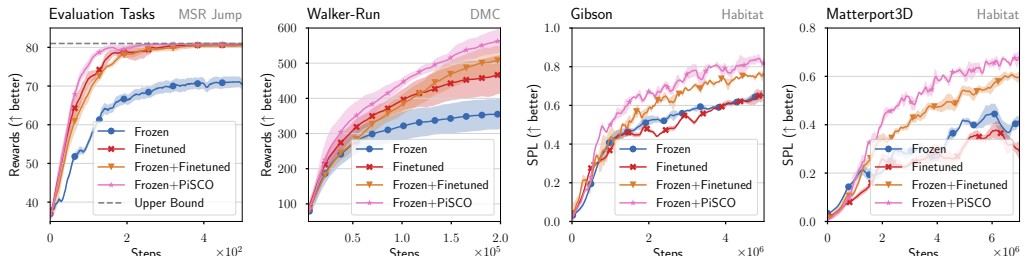

Figure 5: **Partial freezing improves convergence; adding our policy-induced consistency objective improves further.** As suggested by Section 3.5, we freeze the early layers of the feature extractor and finetune the rest of the parameters without (`Frozen+Finetuned`) or with our policy-induced consistency objective (`Frozen+PiSCO`). On challenging tasks (*e.g.*, `Habitat`), partial freezing dramatically boosts downstream performance, while `Frozen+PiSCO` further improves upon `Frozen+Finetuned` across the board.

dissimilarity between the representation of states sampled from $T$:

$$\mathcal{L}^{\texttt{PiSCO}} = \mathop{\mathbb{E}}_{\substack{s \sim \mathcal{B} \\ s_1, s_2 \sim T(\cdot|s)}} \left[ \frac{1}{2} \left( \mathcal{D}(z_1, p_2) + \mathcal{D}(z_2, p_1) \right) \right],$$

where $s_1$ and $s_2$ are two different perturbations obtained from state $s$, and the objective uses a symmetrized version of the dissimilarity measure $\mathcal{D}$. In practice, the `PiSCO` objective is added as an auxiliary term to the underlying RL objective and optimized for the encoder $\phi$ and the projector $h$. Pseudocode is available in Appendix D.

**Remarks**   `PiSCO` is reminiscent of SimSiam (Chen & He, 2020) and only differs in how dissimilarity between embeddings is measured. Our proposed policy-induced similarity measure is crucial for best performance, as shown in Section 5.2. In fact, similar representation learning algorithms for RL could be derived by replacing the embedding similarity measure in existing self-supervised algorithms, such as SimCLR (Chen et al., 2020), MoCo (He et al., 2019), or VICReg (Bardes et al., 2022). We chose SimSiam for its simplicity – no target encoder nor negative samples required – which suits the requirements of RL training.

Alone, `PiSCO` is not a useful objective to solve RL tasks; rather, its utility stems from assisting the underlying RL algorithm in learning representations that are robust to perturbations. *Could we obtain the same benefits by learning the policy with augmented data (and thus side-stepping the need for `PiSCO`)?* In principle, yes. However, the policy objective is typically a function of the rewards which is known to be excessively noisy to robustly learn representations. For example, DrQ avoids learning representations through the policy objective, and instead relies on Bellman error minimization. We hypothesize that `PiSCO` succeeds in learning robust representations because its self-supervised objective is less noisy than the policy objective.

## 5 EXPERIMENTS

We complete our analysis with a closer look at `PiSCO` and partially frozen feature encoders. First, we check whether partial freezing and policy-induced supervision can help accelerate finetuning with RL; second, we compares policy-induced supervision as an RL finetuning objective against more common similarity measures in self-supervised learning.

### 5.1 PARTIAL FREEZING AND POLICY-INDUCED SUPERVISION IMPROVE RL FINETUNING

This section shows that partially freezing a pretrained feature encoder can significantly help stabilize downstream training. We revist the experimental setup of Figure 1, this time including two new transfer variations. The first one is `Frozen+Finetuned`, where we freeze early layers and finetune the remaining ones as with `Finetuned`. The second, `Frozen+PiSCO`, additionally includes the `PiSCO` objective as an auxiliary loss to help finetuning representations. `Frozen+PiSCO` involves one extra hyper-parameter, namely, the auxiliary loss weight, which we set to $0.01$ for all benchmarks.

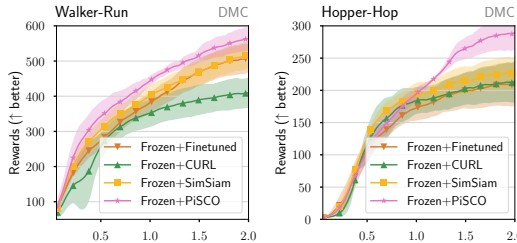

Figure 6: **Policy-induced supervision is a better objective than representation alignment for finetuning.** We compare our policy-induced consistency objective as a measure of representation similarity against popular representation learning objectives (*e.g.* SimSiam, CURL). Policy supervision provides more useful similarities for RL finetuning, which accelerates convergence and reaches higher rewards.

In both cases, we identify which layers to freeze and which to finetune following a similar linear probing setup as for Figure 4a. For each layer, we measure the action value estimation error of pretrained and finetuned representations, and only freeze the first pretrained layers that closely match the finetuned ones. On `MSR Jump`, we freeze up to `Conv3`; on `DeepMind Control` (Walker) up to `Conv2`; and on `Habitat`, we freeze upto `Conv8` for Gibson and upto `Conv7` for Matterport3D. (See the Appendix C for results on `Habitat`.)

We report convergence curves in Figure 5. As suggested by our analysis of Figure 4b, freezing those early layers does not degrade performance; in fact, we see significant gains on `Habitat` both in terms of convergence rate and in asymptotic rewards. Those results are particularly noticeable since `Finetuned` struggled to outperform `Frozen` on those tasks. We also note that `Frozen+PiSCO` improves upon `Frozen+Finetuned` across the board: it accelerates convergence on `MSR Jump`, and also improves asymptotic performance on `DeepMind Control` and `Habitat`. These results tie all our analyses together: they show that (a) freezing early layers can help stabilize transfer in visual RL, (b) which layer to freeze is predicted by how well it encodes action value information, and (c) policy-induced supervision (and `PiSCO`) is a useful objective to accelerate RL finetuning.

## 5.2 POLICY-INDUCED SUPERVISION IMPROVES UPON CONTRASTIVE PREDICTIVE CODING

As a final experiment, we answer how important it is to evaluate similarity through the policy rather than with representation alignment as in contrastive predictive coding. In other words, *could we swap PiSCO for SimSiam and obtain similar results in Figure 5?* Our ablation studies focus on `DeepMind Control` as it is more challenging than `MSR Jump` yet computationally tractable, unlike `Habitat`.

We include `Frozen+SimSiam` and `Frozen+CURL` in addition to `Frozen+Finetuned` and `Frozen+PiSCO`. `Frozen+SimSiam` is akin to `Frozen+PiSCO` but uses the negative cosine similarity to measure the dissimilarity between embeddings $z$ and $p$. For `Frozen+CURL`, we implement the contrastive auxiliary loss of CURL (Laskin et al., 2020), a self-supervised method specifically designed for RL. Both of these alternatives use DrQ as the underlying RL algorithm.

Figure 6 reports convergence curves on `Walker Stand→Run` and `Hopper Stand→Hop`[3]. Including the SimSiam objective typically improves upon vanilla finetuning, but including CURL does not. We hypothesize that most of the benefits from CURL are already captured by DrQ, as shown in prior work (Yarats et al., 2021). `PiSCO` significantly outperforms the standard self-supervised methods thus demonstrating the efficacy of using the policy when measuring similarity between embeddings. We hypothesize that representation alignment objectives naturally map two similarly looking states to similar representations, even when they shouldn't (*e.g.*, when the agent is 15 and 14 pixels away from the obstacle in `MSR Jump`). Instead, `PiSCO` is free to assign different representations since those two states might induce different actions ("move right" and "jump", respectively).

## 6 CONCLUSION

Our analysis of representation learning for transfer reveals several new insights on the roles of finetuning: similarity between representations should reflect whether they induce the same or similar distribution of actions; not all layers in encoders of states need to be adapted. Building on those insights, we develop a hybrid approach which partially freezes the bottom (and readily transferrable) layers while finetuning the top layers with a policy-induced self-supervised objective. This approach is especially effective for hard downstream tasks, as it alleviates the challenges of finetuning rich visual representations with reinforcement learning.

---

[3]`Hopper-Hop` is especially difficult; *e.g.*, DrQ-v2 fails to solve it after 3M iterations (Yarats et al., 2022).

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

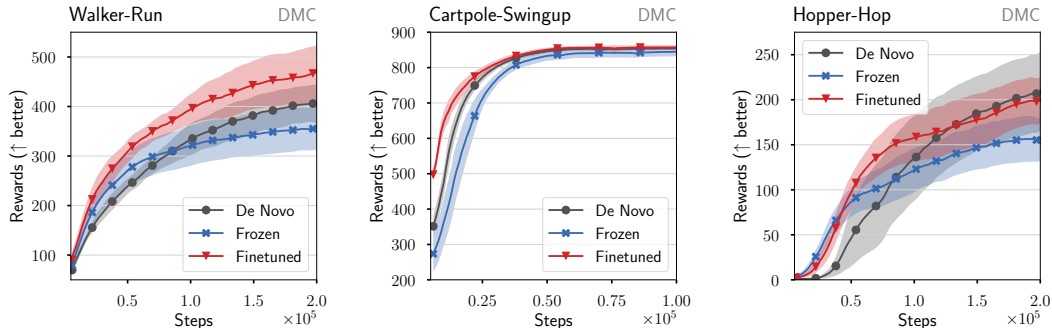

Figure 7: **When should we freeze or finetune pretrained representations in visual RL?** Reward transfer curves on `Cartpole` and `Hopper DeepMind Control` domains; replicates Figure 1. Freezing pretrained representations can underperform no pretraining at all (Figures 1a and 1b) or outperform it (Figures 1c and 1d). Finetuning representations is always competitive, but fails to significantly outperform freezing on visually complex domains (Figures 1c and 1d). Solid lines indicate mean over 5 random seeds, shades denote 95% confidence interval. See Section 3.2 for details.

## A    APPENDIX

## B    TESTBED DETAILS

This section details the pretraining and transfer setups for the `MSR Jump`, `DeepMind Control`, and `Habitat` domains.

### B.1    `MSR Jump` (TACHET DES COMBES ET AL., 2018)

As discussed in Section 3, the goal of `MSR Jump` tasks is for the agent to reach the right-hand side of the screen. The agent always starts on the left-hand side and has to jump over a single gray obstacle. The agent observes the entire screen from pixel values, and the only factors of variation are the $x$ and $y$ coordinates of the obstacle (*i.e.*, its horizontal position and the floor height). Its only possible actions are to move right or jump. The jump dynamics are shared across tasks and mid-air actions don't affect the subsequent states until the agent has landed (*i.e.*, all jumps go up by 15 pixels diagonally, and down by 15 pixels diagonally). Our experiments carefully replicate the reward and dynamics of Tachet des Combes et al. (2018), save for one aspect: we increase the screen size from 64 pixels to 84 pixels, as this significantly accelerated training with the DQN (Mnih et al., 2015) feature encoder. We provide a fast GPU-accelerated implementation of `MSR Jump` tasks in our code release (see Appendix E).

#### B.1.1    PRETRAINING SETUP

**Source tasks**    We pretrain on 70 source tasks, varying both the obstacle $x$ (from 15 to 50 pixels) and $y$ (from 5 to 50) coordinates, every 5 pixel. We don't use frame stacking nor action repeats.

**Learning agent**    We train with PPO (Schulman et al., 2017) for 500 iterations, where each iteration collects 80 episodes from randomly sampled source tasks. We use those episodes to update the agent 320 times with a batch size of 64, a discount factor of 0.99, and policy and value clip of 0.2. The learning rate starts at 0.0005 and is decayed by 0.99 once every 320 update. The agent consists of the 4-layer DQN feature encoder and linear policy and value heads. All models use GELU (Hendrycks et al., 2016) activations.

#### B.1.2    TRANSFER SETUP

**Downstream tasks**    At transfer time, we replace the 70 source tasks with 70 unseen downstream tasks. Those downstream tasks are obtained by shifting the obstacle position ($x$ and $y$ coordinates) of each source task by 2 pixels. For example, if a source task has an obstacle at $x = 20$ and $y = 30$ there is a corresponding downstream task at $x = 22$ and $y = 32$.

**Learning agent**   The learning agent is identical to the source tasks, but rather than randomly initializing the weights of the feature encoder we use the final weights from pretraining. As mentioned in the main text, we also introduce one additional hyper-parameter as we decouple the learning rate for the feature encoder and policy / value heads. For `Finetuned`, these learning rates are $0.001$ for the heads and $0.0001$ for the feature encoder; both learning rates are decayed during finetuning.

### B.2   `DeepMind Control` (TASSA ET AL., 2018)

The `DeepMind Control` tasks build on top of the MuJoCo physiscs engine (Todorov et al., 2012) and implement a suite of robotic domains. Our experiments focus on `Walker`, `Cartpole`, and `Hopper` (see Appendix C.2 for the latter two). `Walker` and `Hopper` are robot locomotion domains, while `Cartpole` is the classic cartpole environment. On all domains, the agent receives an agent-centric visual $84 \times 84$ RGB observation, and controls its body with continuous actions. Our task implementation exactly replicate the one from DrQ (Yarats et al., 2021).

#### B.2.1   PRETRAINING SETUP

**Source task**   On `Walker`, the source task is `Walk`, where the agent is tasked to walk in the forward direction. On `Cartpole` it is `Balance`, where the pole starts in a standing position and the goal is to balance it. And on `Hopper` it is `Stand`, where the agent needs to reach a (still) standing position. For all tasks, we keep a stack of the last 3 images and treat those as observations.

**Learning agent**   Our learning agent is largely inspired by DrQ-v2 (Yarats et al., 2022), and comprises a 4-layer convolutional feature encoder, a policy head, and a twin action value head. Both the policy and action value heads consist of a projection layer, followed by LayerNorm (Ba et al., 2016) normalization, and a 2-layer MLP with $64$ hidden units and GELU (Hendrycks et al., 2016) activations. We train with DrQ-v2 for 200k iterations, starting the policy standard deviation at $1.0$, decaying it by $0.999954$ every iteration until it reaches $0.1$. We use a $512$ batch size and don't use n-steps bootstrapping nor delayed updates. For pretraining, we always a learning rate of $3e^{-4}$ and default Adam (Kingma & Ba, 2014) hyper-parameters.

#### B.2.2   TRANSFER SETUP

**Downstream task**   On `Walker`, we transfer to `Run`, where the agent needs to run in the forward direction. On `Cartpole`, the downstream task is `Swingup`, where the pole starts in downward position and should be swung and balanced upright. On `Hopper`, we use `Hop`, where the agents move in the forward direction by hopping.

**Learning agent**   We use the same learning agent as for pretraining, but initialize the feature encoder with the weights obtained from pretraining. We also add a learning rate hyper-parameter when finetuning the feature encoder, which is tuned independently for each setting and method.

### B.3   `Habitat` (SAVVA ET AL., 2019)

On `Habitat`, the goal of the agent is to navigate indoor environments to reach a desired location. To solve this point navigation task, it relies on two sensory inputs: a GPS with compass and visual observations. The visual observations are $256 \times 256$ RGB images (*n.b.*, without depth information). Our codebase builds on the task and agent definitions of "Habitat-Lab" (Szot et al., 2021), and only modifies them to implement our proposed methods.

#### B.3.1   PRETRAINING SETUP

**Source task**   We use the ImageNet (Deng et al., 2009) dataset as source task to pretrain our feature encoder.

**Learning agent**   The learning agent is a ConvNeXt-tiny (Liu et al., 2022) classifier, trained to discriminate between ImageNet-1K classes. We directly use the pretrained weights provided by the ConvNeXt authors, available at: `https://github.com/facebookresearch/ConvNeXt`

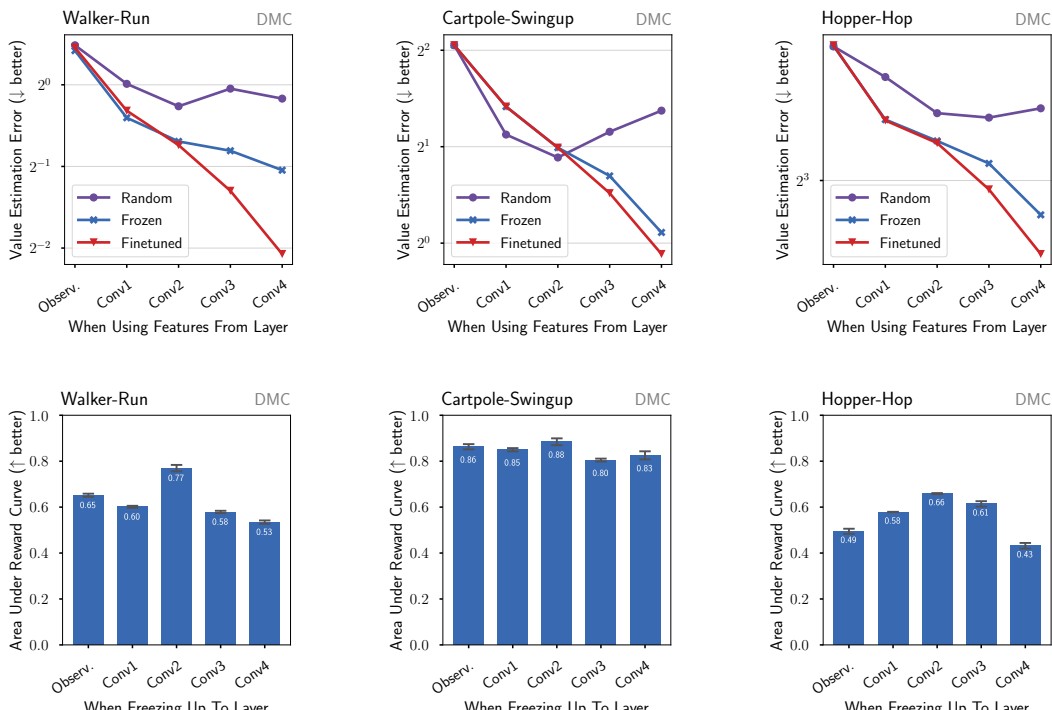

Figure 8: **Frozen layers that retain the same information as finetuned layers can be kept frozen.** Replicates Figure 4a on `Cartpole` and `Hopper` `DeepMind Control` domains. **(top)** Mean squared error when linearly regressing action values from representations computed by a given layer. Some early layers are equally good at predicting action values before or after finetuning. This suggests those layers can be kept frozen, potentially stabilizing training. See Section 3.5 for details. **(bottom)** Area under the reward curve when freezing up to a given layer and finetuning the rest. Freezing early layers does not degrade performance, and sometimes improves it (*e.g.*, up to `Conv2` on `DeepMind Control`).

### B.3.2 TRANSFER SETUP

**Downstream tasks**   For transfer, we load maps from the Gibson (Xia et al., 2018) and Matterport3D (Chang et al., 2017) datasets. Both are freely available for academic, non-commercial use at:

- Gibson: `https://github.com/StanfordVL/GibsonEnv/`
- Matterport3D: `https://github.com/niessner/Matterport/`

**Learning agent**   We take the pretrained ConvNeXt classifier, discard the linear classification head, and only keep its trunk as our feature encoder. We train a 2-layer LSTM policy head and a linear value head on top of this feature encoder with DDPPO (Wijmans et al., 2020) for 5M iterations on Gibson and 10M on Matterport3D. For both tasks, we collect $128$ steps with $4$ workers for every update, which consists of 2 PPO epochs with a batch size of $4$. We set PPO's entropy coefficient to $0.01$, the value loss weight to $0.5$, GAE's $\tau$ to $0.95$, the discount factor to $0.99$, and clip the gradient norm to have $0.2$ $l_2$-norm. The learning rate for the policy and value heads is set to $2.4e^{-4}$, and the feature encoder's learning rate to $1e^{-4}$ for `Frozen+Finetuned`. For `Frozen+Varnish`, we use a single learning rate set to $1e^{-4}$ for the feature encoder and policy / value heads (but $2.5e^{-4}$ gave similar results). We never decay learning rates.

## C   ADDITIONAL EXPERIMENTS

This section provides additional details on the experiments from the main text, and also includes additional results to further support our conclusions.

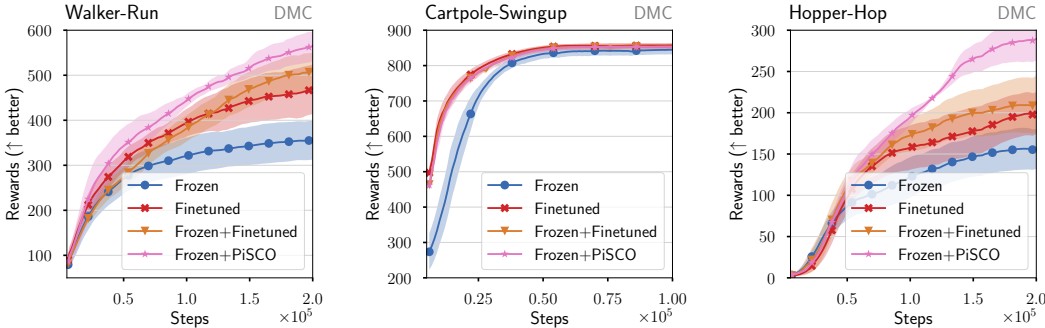

Figure 9: **Partial freezing improves convergence; adding our policy-induced consistency objective improves further.** As suggested by Section 3.5, we freeze the early layers of the feature extractor and finetune the rest of the parameters without (`Frozen+Finetuned`) or with our policy-induced consistency objective (`Frozen+PiSCO`). On challenging tasks (*e.g.*, `Hopper`), partial freezing dramatically boosts downstream performance, while `Frozen+PiSCO` further improves upon `Frozen+Finetuned` across the board.

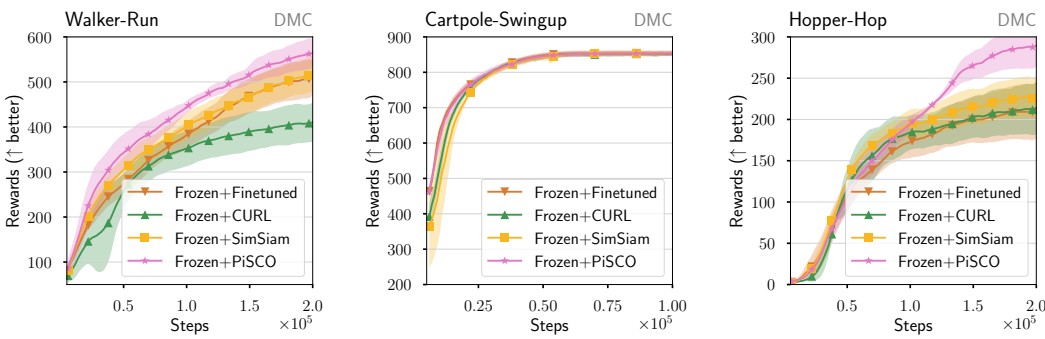

Figure 10: **Policy-induced supervision is a better objective than representation alignment for finetuning.** We compare our policy-induced consistency objective as a measure of representation similarity against popular representation learning objectives (*e.g.* SimSiam, CURL). Policy supervision provides more useful similarities for RL finetuning, which accelerates convergence and reaches higher rewards.

## C.1  COMPUTING ACTION VALUES FOR THE LAYER-BY-LAYER LINEAR PROBING EXPERIMENTS

We follow a similar protocol for all tasks to obtain action values for layer-by-layer linear probing experiments. As stated in Section 3.5, we first collect 10,000 pairs using a policy trained (or finetuned) on the downstream task. We then compute the returns for each state-action pair by discounting the rewards observed in the collected trajectories. Then, we fit a 64-hidden unit 2-layer MLP with GELU activations to predict action values from the collected state-action pairs, and store the predicted action-values. On `MSR Jump` and `Habitat` we use the one-hot encoding of the actions rather than their categorical values. Finally, we use the MLP-predicted action values as regression targets for our linear probing models.

## C.2  FULL RESULTS ON DeepMind Control

As mentioned in Section 3.1, we also ran our analysis on two `DeepMind Control` domains other than `Walker`: `Cartpole` and `Hopper`. Both tasks use the same pretraining and transfer setups as `Walker`. In `Cartpole`, we pretrain on the `Balance` task, where the pole starts in an upright position and our goal is to keep it blanced. We then transfer to `Swingup`, where the pole starts upside down and the goal is to swing and balance it upright. With `Hopper`, the pretraining task is `Stand` where the agent is asked to stand up still; the downstream task is `Hop` where the agent should move in the forward directino as fast as it can. Note that `Hopper-Hop` is considered more challenging than `Cartpole` and `Walker` tasks, and DrQ-v2 fails to solve it even after 3M iterations (Yarats et al., 2022).

For both domains, all analysis results are qualitatively similar to the `Walker` ones from the main text. Figure 7 shows once more that freezing representations is not sufficient for transfer. Figure 8

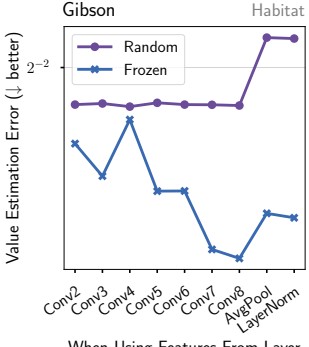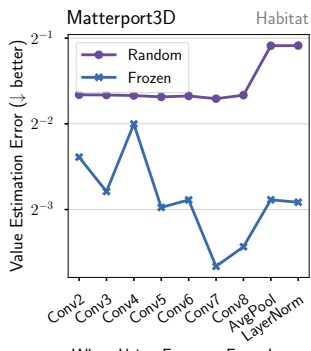

Figure 11: **Identifying which layers to freeze on Habitat tasks.** We replicate the layer-by-layer linear probing experiments on Habitat with the ImageNet pretrained feature extractor. Although the downward trend is less evident than in Figure 4a, we clearly see that layers `Conv8` and `Conv7` yield lowest value prediction error on Gibson and Matterport3D, respectively.

replicates Figure 4a. On both `Cartpole` and `Hopper`, freezing beyond layer `Conv2` increases the value estimation error (top panels); conversely, freezing up to those layers yields the best AURC (bottom panels). Note that on `Cartpole` the degradation in value error is minimal and so is the difference in AURC. We see in Figure 9 that partially freezing the feature encoder yields better transfer results, especially when combined with `PiSCO`. Finally, Figure 10 reconfirms our hypothesis that a policy-driven similarity metric improves RL finetuning.

### C.3   SELECTING FROZEN LAYERS IN `Habitat`

We decided to freeze up to layer `Conv8` and `Conv7` for the Gibson and Matterport3D experiments in Section 5.1. This section explain how our analysis informs the choice of those layers, without resorting to a finetuned policy nor layer-by-layer freezing experiments as in Figure 4b.

Building on our analysis, we obtained $10,000$ state-action pairs from a policy trained directly on Gibson and Matterport3D. This policy is provided by "Habitat-lab" but in practice a couple of expert demonstrations might suffice. As in Figure 4a, we fit an action value function and save the action values for the observation-action pairs from the collected state-action pairs. Then, we replicate the layer-by-layer linear probing experiment of Section 3.5 with a randomly initialized feature encoder and the one pretrained on ImageNet. Two differences from Section 3.5 stem from the large and visually rich observations on `Habitat`. First, we project each layer's representations to a PCA embedding of dimensions $600$ rather than $50$[4]; second, we omit the first convolutional layer as PCA is prohibitively slow on such large representations.

Figure 11 displays those results. As expected, the downward trend is less evident than in Figure 4a since the transfer gap between ImageNet and `Habitat` is larger than between tasks in `MSR Jump` or `DeepMind Control`. Nonetheless, we can easily identify the representations in layers `Conv8` and `Conv7` as the most predictive of action value on Gibson and Matterport3D, thus justifying our choice.

### C.4   `DeepMind Control` TRANSFER FROM IMAGENET

Given the promising results of large feature extractors pretrained on large amounts of data Figures 1c and 1d, we investigate whether similarly trained feature extractors would also improve transfer on `DeepMind Control` tasks. To that end, we use the same ConvNeXt as on our `Habitat` experiments, freeze its weights, learn policy and action-value heads on each one of the three `DeepMind Control` tasks (replicating the setup of Figure 1).

---

[4]Dimensions ranging from 500 to 750 work equally well. Smaller dimensions yield underfitting as PCA doesn't retain enough information from the representations. Larger dimensions yield overfitting and pretrained representations don't do much better than random ones.

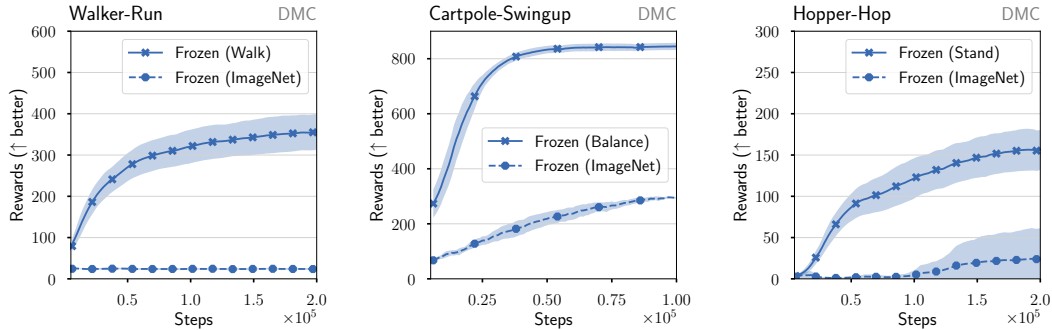

Figure 12: **Pretraining on large and diverse data can hurt transfer when the generalization gap is too large.** When transferring representations that are pretrained on ImageNet to `DeepMind Control` tasks, we see a signficant decrease in convergence rate. We hypothesize this is due to the lack of visual similarity between ImageNet and `DeepMind Control`.

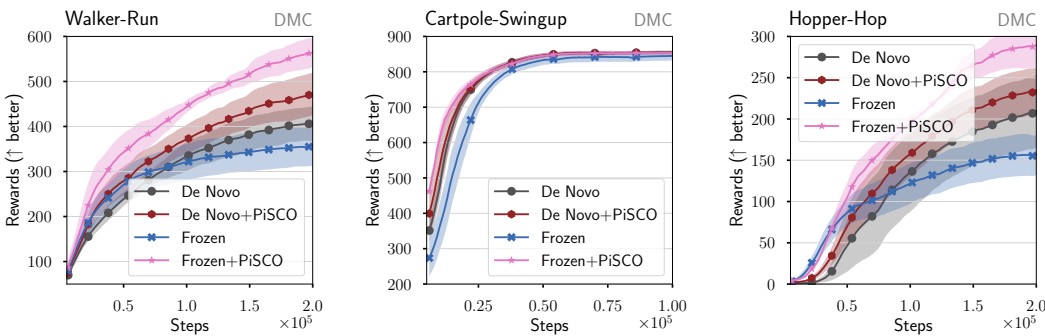

Figure 13: `PiSCO` **can also improves representation learning on source tasks but shines with pretrained representations.** When benchmarking `PiSCO` combined with fully finetunable features (`De Novo+PiSCO`), we observe that it marginally outperforms DrQ-v2 on the downstream tasks (`De Novo`). However, the best performance is obtained when also transferring task-agnostic features (`Frozen+PiSCO`).

Figure 12 provides convergence curves comparing the 4-layer CNN pretrained on the source `Deep-Mind Control` task (*i.e.*, either `Walker-Walk`, `Cartpole-Balance`, or `Hopper-Stand`) against the ConvNeXt pretrained on ImageNet. The CNN drastically outperforms the ConvNeXt despite having fewer parameters and trained on less data. We explain those results as follows. The smaller network is trained on data that is more relevant to the downstream task, thus has a smaller generalization gap to bridge. In other words, more data is useful insofar as it is relevant to the downstream task. More parameters might help (*i.e.*, pretraining the ConvNeXt on `DeepMind Control` tasks) but it is notoriously difficult to train very deep architectures from scratch with reinforcement learning, as shown with `De Novo` on `Habitat` tasks.

## C.5 `De Novo` FINETUNING WITH `PiSCO`

As an additional ablation, we investigate the ability of `PiSCO` to improve reinforcement learning from scratch. In Figure 13, we compare the benefits of using `PiSCO` with when representations are (partly) frozen *v.s.* fully finetunable (`De Novo`). We see that `PiSCO` always improves upon learning from scratch (`De Novo+PiSCO` outperforms `De Novo`) but that it is most beneficial when the task-agnostic features are frozen (`Frozen+PiSCO` outperforms `De Novo+PiSCO`).

## C.6 `PiSCO` WITHOUT PROJECTION LAYERS

The original SimSiam formulation includes a projector layer $h(\cdot)$, and Chen & He (2020) show that this projector is crucial to prevent representation collapse. *Does this also holds true for RL?*

To answer this question, we compare the performance of a (partially) frozen feature extractor finetuned with `PiSCO`, with and without a projection layer, in Figure 14. The results clearly answer our question

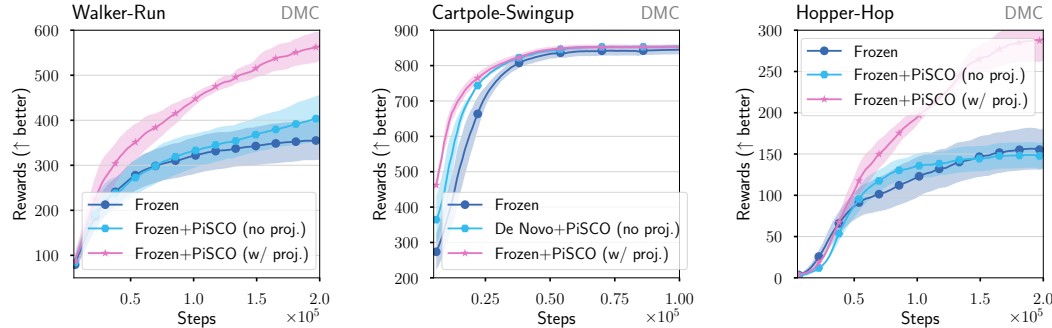

Figure 14: **Is the projector $h(\cdot)$ necessary in `PiSCO`'s formulation?** Yes. Removing projection layer when finetuning task-specific layers (and freezing task-agnostic layers) drastically degrades performance on all `DeepMind Control` tasks.

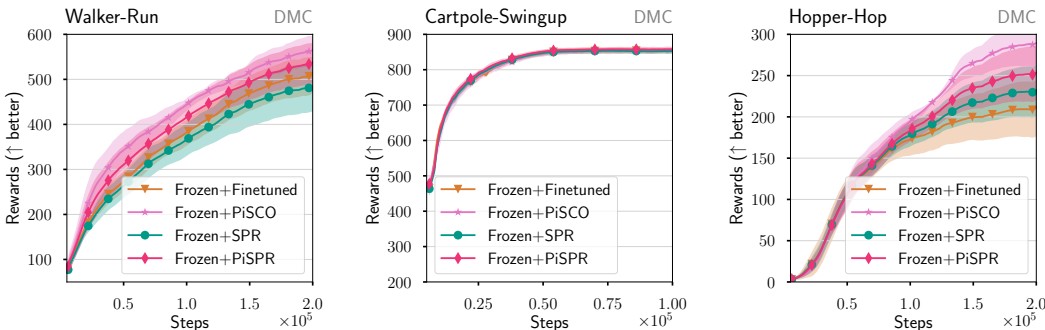

Figure 15: **Combining `SPR` with `PiSCO` significantly improves the performance of `SPR`.** Swapping the cosine similarity objective in SPR (Schwarzer et al., 2021) for the policy-induced objective suggested by our analysis significantly improves finetuning. Still, finetuning with `PiSCO` (based on SimSiam (Chen & He, 2020)) yields the best performance, while remaining easier to implement and faster in terms of wall-clock time.

in the affirmative: the projector is also required and removing it drastically degrades performance on the downstream task.

## C.7 COMPARING AND COMBINING WITH SPR

To illustrate future avenues for extending the ideas motivating `PiSCO`, we combine them with the self-predictive representation (SPR) objective of Schwarzer et al. (2021). We replace SimSiam for SPR as the auxiliary self-supervised objective used during finetuning. Concretely, reimplemented SPR for `DeepMind Control` tasks and compare the original implementation (which uses cosine similarity $\cos(p_1, p_2)$ to compare representations) against a variant (`PiSPR`) which uses the policy-induced objective $\frac{1}{2}\left(\mathcal{D}(p_1, p_2)) + \mathcal{D}(p_2, p_1)\right)$[5].

In terms of implementation details, we found that setting the number of contiguous steps sampled from the replay buffer to $K = 3$ yielded the best results on `DeepMind Control` tasks. The transition model is a 2-layer MLP (1024 hidden units) with layer normalization and ReLU activations. We used the same projector as as for SimSiam (which does not reduce dimensionality), and a linear predictor with layer-normalized inputs.

Figure 15 reports the results for partially frozen feature extractors with different finetuning objectives. We observe that using SPR typically performs as well as finetuning the task-specific layers with the RL objective. In that sense, finetuning with `PiSCO` offers an compelling alternative: it is substantially cheaper (3x faster in wall-clock time), doesn't require learning a transition model nor keeping track of momentum models, and is simpler to tune (fewer hyper-parameters). More interestingly, we find that replacing the cosine similarity in SPR for the policy-induced objective (*i.e.*, `PiSPR`) significantly improves performance over SPR. Those results further validate the generality of our analysis insights.

---

[5]Note: $p_1$ and $p_2$ correspond to $\hat{y}_{t+k}$ and $\tilde{y}_{t+k}$ in the original SPR notation.

# D PSEUDOCODE

Here we provide pseudocode for implementing PiSCO on top of two popular reinforcement learning algorithms. Algorithm 1 adds PiSCO to PPO (Schulman et al., 2017), while Algorithm 2 adds it to DrQ-v2 (Yarats et al., 2022).

---

**Algorithm 1** PPO with PiSCO

---

```
1   # sample transitions from replay
2   s, a, r, s′ = replay.sample(batch_size)
3   z = ϕ(s)
4
5   # compute value loss
6   ℒ^V = 0.5 · (V(z) - discount(r))²
7
8   # compute policy loss
9   Δ_π = log π(a | z) − log π_old(a | z)
10  A = GAE(V(z), r, γ, τ)
11  ℒ^π = min(exp(Δ_π) · A, clip(exp(Δ_π) · A, 1 − ϵ, 1 + ϵ)
12
13  # compute PiSCO loss
14  z₁, z₂ = ϕ(data_augment(s)), ϕ(data_augment(s))
15  p₁, p₂ = h(z₁), h(z₂)
16  ℒ^PiSCO = 0.5 · KL(⊥(π(· | z₁))‖π(· | p₂)) + 0.5 · KL(⊥(π(· | z₂))‖π(· | p₁))
17
18  adam.optimize(ℒ^π + ν · ℒ^V + λ · ℒ^PiSCO − β · H(π(· | z)))   # optimizes V, π, h, and ϕ
```

---

---

**Algorithm 2** DrQ-v2 with PiSCO

---

```
1   # sample transitions from replay
2   s, a, r, s′ = replay.sample(batch_size, n_steps, γ)
3
4   # compute policy loss
5   z = ⊥(ϕ(data_augment(s)))
6   â = π(· | z).rsample()
7   ℒ^π = − min(Q₁(z, â), Q₂(z, â))
8   adam.optimize(ℒ^π)   # only optimizes π
9
10  # compute action-value loss
11  z = ϕ(data_augment(s))
12  z′ = ϕ(data_augment(s′))
13  a′ = π(· | z′).sample()
14  q₁, q₂ = Q₁(z, a), Q₂(z, a)
15  q′ = ⊥(r + γ · min(Q₁(z′, a′), Q₂(z′, a′)))
16  ℒ^Q = 0.5 · (q₁ − q′)² + 0.5 · (q₂ − q′)²
17
18  # compute PiSCO loss
19  z₁, z₂ = ϕ(data_augment(s)), ϕ(data_augment(s))
20  p₁, p₂ = h(z₁), h(z₂)
21  ℒ^PiSCO = 0.5 · KL(⊥(π(· | z₁))‖π(· | p₂)) + 0.5 · KL(⊥(π(· | z₂))‖π(· | p₁))
22
23  adam.optimize(ℒ^Q + λ · ℒ^PiSCO)   # only optimizes Q₁, Q₂, h, and ϕ
```

---

# E CODE RELEASE

Together with our submission, we release example code to implement PiSCO in the code/ directory of the Supp. Material. We plan to publicly release complete implementations, together with examples for the pretrain-and-transfer pipeline. We describe the main files below.

- `msr_jump.py`: our GPU implementation of MSR Jump tasks.
- `ppo_pisco.py`: our PyTorch (Paszke et al., 2019) implementation of PiSCO on top of PPO.
- `drqv2_pisco.py`: our PyTorch (Paszke et al., 2019) implementation of PiSCO on top of DrQ-v2.

