# OpenReview forum: "Policy-Induced Self-Supervision Improves Representation Finetuning in Visual RL"
_ICLR.cc/2023/Conference — Submitted to ICLR 2023_

### Official Review · Reviewer_b3ER · 2022-10-17

**Confidence:** 3
**Correctness:** 3
**Technical Novelty And Significance:** 3
**Empirical Novelty And Significance:** 2
**Recommendation:** 6

**Clarity, Quality, Novelty And Reproducibility:**

Clarity: the writing is easy to follow, but the high-level clarity is a bit lacking (see the "Weaknesses" section).

Novelty: to the best of my knowledge, the contributions are novel.

Reproducibility: the authors provide both algorithm pseudocode and their actual code in the supplementary material. The paper contains many experimental details.


**Strength And Weaknesses:**

Strengths:
1. The proposed objective, PiSCO, seems elegant and promising. I am curious if the authors could say more about the projector $h(\cdot)$ that is used in the PiSCO loss. The intuition for PiSCO makes sense, but I am not sure why why should compare $\pi(\cdot|h_1)$ and $\pi(\cdot|z_2)$ instead of simply $\pi(\cdot|z_1)$ and $\pi(\cdot|z_2)$?
2. The paper is well-written, and the authors attempt to be insightful in their discussion.

Weaknesses:
1. The narrative of the paper feels a bit incoherent. It feels to me that the freezing vs finetuning investigation did not well-motivate PiSCO, and that PiSCO does not address the original problem statement that motivated said investigation. From what I can tell, PiSCO is an auxiliary loss to be used during RL, not a pretraining objective. I think this paper's contributions would be more clear if either: (a) the authors proposed a pretraining objective, or (b) the authors more directly motivated PiSCO, e.g. by discussing robustness or data augmentation/efficiency.
2. Related to the previous point, if the authors were to consider option (b), then I feel that PiSCO could benefit from a more thorough experimental evaluation (a greater variety of tasks/domains, with more emphasis paid to sample efficiency).
3. The authors make broad statements about finetuning vs freezing solely based on results from a toy environment, MSR jump. I understand the attractiveness of that domain (in that the optimal policy is known), but I'm wondering if it would be possible to conduct a similar study in other domains as well?



**Summary Of The Paper:**

The authors compare finetuning and freezing pretrained representations on various RL tasks.
They seek to understand under what conditions each of these can be effective, and when one should be preferred to the other.
On a simple task where the optimal policy is known, they show that the pretrained representation does contain enough information to regress the optimal actions, but finetuning still fails to recover the optimal policy in spite of this.
They use insights from these experiments to propose a new self-supervised objective for RL, and evaluate this objective on a variety of tasks.


**Summary Of The Review:**

I think that the proposed approach shows promise, but the motivation is unclear (mainly per Weaknesses #1). I would feel comfortable accepting this paper if this were addressed (and the experiments/evaluation may need to change accordingly).

---

> ### Author Response · Authors · 2022-11-15
> **Response to Reviewer b3ER**
>
> Thank you for raising some concerns on the motivation (and for the encouraging words regarding PiSCO). We’ve addressed some of your concerns below, point-by-point.
>
>
> * What is the intuition for the projector h(z)? Could we use pi(a | z) instead of pi(a | p)?
>    * The projector comes directly from the SimSiam formulation, and matters to get the best performance (both in their paper and ours). One intuition provided by the authors is that this projector helps prevent representation collapse. Due to your and Reviewer nmYq’s interest, we’ve added Figure 14 ablating the projector in Appendix C.6. The results clearly show that the projector is needed: without it, Frozen+PiSCO barely outperforms freezing all layers before transfer.
>
>
> * The writing is incoherent: how does PiSCO address the original problem statement, and how does the analysis motivate PiSCO?
>    * PiSCO directly addresses the problem of unstable finetuning shown in Fig. 1.c-d: it drastically stabilizes finetuning by providing a learning signal for the feature extractor that is less susceptible to the noise in the early stages of finetuning (Fig. 5.c-d). That’s because PiSCO doesn’t rely on rewards but rather ensures policy consistency for two versions of the same state. The idea of the policy being consistent under data augmentation is a direct consequence of our analysis: we show that Finetuned representations yield purer clusters (→ policy consistency; see Fig. 3.b) and are more robust to noise (→ consistency w.r.t. augmentations; see Fig. 3.c).
>    * In terms of coherence, the paper follows this structure:
>       1. Illustrate issues with freezing and finetuning representations.
>          * Freezing issues: can perform poorly on downstream tasks, even when they are very similar to source tasks (Fig. 1.a-b).
>          * Finetuning issues: can be too unstable for visually complex tasks, failing to outperform freezing (Fig. 1.c-d).
>       2. Study why finetuning works well.
>          * It works because finetuned representations yield more consistent policies (Fig. 3.b) and they are more robust to noise (Fig. 3.c).
>       3. Study if we can avoid finetuning all layers, and identify which ones require finetuning.
>          * Yes, we can avoid finetuning early layers; those layers can be identified with a simple linear probing experiment.
>       4. Propose PiSCO, a method that builds on “why finetuning works”.
>          * Namely, combines the ideas of policy consistency and robustness to noise.
>       5. Validate our analyses and proposed methodology.
>
>
> * The paper would be more clear if we proposed a) a pretraining objective, or b) directly motivate PiSCO from robustness / data augmentation / sample efficiency perspective.
>    * Thanks for the suggestions! We seriously considered both alternatives and arrived at the following conclusions.
>    * As foreseen by Reviewer b3ER, option a) is not satisfying because it sidesteps the issue of representation finetuning with an impractical solution; namely, changing the pretraining objective. This is undesirable because it is much more convenient for practitioners to download off-the-shelve pretrained weights (e.g., trained on ImageNet as in our Habitat experiments) rather than pretraining the weights themselves.
>    * Option b) is more appealing because it directly ties into our setting of study, and we could change the motivation from “the challenges of freezing / finetuning in RL” to improving robustness or sample efficiency in RL. But, we see our discussion and analysis of freezing / finetuning as a central and novel contribution of our work. In fact, one of our goals is to shed light on these under-appreciated issues in RL transfer. We hope our work can motivate further study in the robustness and efficiency properties of representations finetuned with SSL objectives (like PiSCO) – we too are curious of the tradeoffs of adapting representations without rewards.
>
>
> * Some statements are too broad and only supported by results on MSR Jump. Can we validate them on other tasks?
>    * Only the results in Fig. 3 are validated exclusively on MSR Jump. This is because jump tasks exhibit *2* special properties: 1. an optimal policy is known and easily written by hand, and 2. the pretrained representations are good enough for perfectly solving the downstream tasks (at least on paper, see Sec. 3.2). We don’t know of another set of tasks where this is true.
>
>
> * PiSCO would benefit from more thorough evaluations.
>    * In addition to the 6 different transfer settings (Jump, 3x DMC, 2x Habitat) and the ablation studies in Fig. 6 and 10, we have added the following experiments during the rebuttal period:
>       * Appendix C4: Using pretrained feature extractor trained on more data for freezing baselines.
>       * Appendix C5: Combining PiSCO with De Novo learning.
>       * Appendix C6: Ablating the projection layer in PiSCO.
>       * Appendix C7: Comparing and combining PiSCO with SPR.

---

> > ### Comment · Reviewer_b3ER · 2022-12-06
> > **Response to Rebuttal**
> >
> > Thank you for the detailed rebuttal and the additional experiments. Considering these, and seeing the discussion with the other reviewers, I am increasing my score from 5 to 6.

---

### Official Review · Reviewer_nmYq · 2022-10-23

**Confidence:** 4
**Correctness:** 3
**Technical Novelty And Significance:** 3
**Empirical Novelty And Significance:** 2
**Recommendation:** 6

**Clarity, Quality, Novelty And Reproducibility:**

The paper is very clearly written and easy to read.
I believe most technical details required to allow reproduction are provided, with one exception: I could not find specifications on the data augmentation used.



**Strength And Weaknesses:**

All 3 contributions outlined above are more or less independent from each other, as such this section will be divided into three parts discussing the strength and weaknesses of each of them.

## Contribution 1

The paper starts by analyzing the performance of two main transfer strategies, namely freezing the representations or finetuning them. This is compared to the baseline of training the representations from scratch on the target task (strategy nicknamed "de novo"). In a nutshell, the results can be summarized as follows:
- For the toy jump task as well as the Deepmind control suite, finetuning outperforms "de novo" while freezing representations substantially decreases performance
- For the Habitat tasks, both strategies outperform "de novo" but none seems strictly better than the other.

The paper provides a thorough and solid analysis of the failure of the freezing of the representations for the jump task, linking it to robustness issues.
However, in my opinion, this falls short of making a broader conclusion: the results above suggest a meaningful qualitative difference between the Habitat task and the other ones (in Habitat, freezing works while it doesn't in the others). The reason for this difference is not discussed in the paper, yet it seems to be somewhat at the crux of the matter. Does this difference comes from intrinsic characteristics of the environments or from other factors?
As for the other factors, we can list 2 main ones:
- Network capacity/architecture: The jump task uses the original DQN network, which is rather primitive (no skip connections for example) and has a tiny capacity (around 100k trainable parameters). By contrast, the Habitat task uses a state-of-the art CNN, with 28M parameters, ie two orders of magnitude larger.
- Pre-training tasks: The convnext for Habitat is pre-trained on ImageNet, while in the other experiments the backbones are pretrained on related control tasks.

In a nutshell, the networks used for Habitat have received a much more varied pre-training and have a much higher capacity. It is well known in the vision community (see [1] for example) that both these properties improve robustness and transferability of the features, making the results presented in the paper somewhat unsurprising.

Overall, it is not clear to me if the "frozen" transfer strategy would still fail in the two tasks where it currently does, if the same convnext, with the same pretraining, was used as the feature extractor. Depending on the answer to that question, a slightly different conclusion could be made: similarly to trends in other domains (Vision, NLP, speech, ...) does RL benefit from larger models pre-trained on large, general datasources?

Finally, I believe the paper could discuss a bit the alternate method to full freezing and finetuning that is being privileged more and more both for vision and NLP: freezing the pre-trained weights but adding a small amount of trainable parameters to allow "adpatation" of the learned representation to the new task. See [2], [3] and references therein.
Another thread of work of interest is the idea of using a different learning rate per layer (typically lower for the deepest layers and increasing from there). See [4] for an example.


## Contribution 2

The paper proposes to resort to linear probing of the feature of each intermediate layer to determine where the cut-off between frozen and fine-tuned should be made.
While the method seems sound, several design choices are not really explained:
- What is the justification behind reducing the dimensionality with PCA first, as opposed to simply using a linear layer directly on the features?
- The paper states that the probing is using the output of a action-value function network. Would it be possible to alleviate the need to train such a model by simply predicting an empirical estimate (eg the mean) of the expected discounted return in each state?


Figure 4.b (and similar figures in the appendix) would benefit from error bars since it's not clear what the variance is. Why does Fig. 11 not contain the estimates for the fine-tuned features?

Finally, the main weakness of the approach is its computational cost. Section 5.1 states "For each layer, we measure the action value estimation error of pretrained and finetuned representations, and only freeze the first pretrained layers that closely match the finetuned one". This implies access to the said fine-tuned representations, which in-turn implies that a full RL training is required as a first step. As such, this doubles the cost of the method (both in wall-clock time as well as sample complexity), significantly reducing its practicality. For proper comparison, it would be necessary to see whether the "Finetuned" baseline from figure 5 is still lower than the "Frozen+Finetuned" if the former is trained for twice the number of steps.
Perhaps, a "softer" scheme akin to [4] would do the job equally well while side-stepping the computational cost entirely.

## Contribution 3

The last contribution is a self-supervised auxiliary objective aiming at improving the quality of the representation. Though it is inspired from findings from contribution #1, it is a contribution that is not specific to the transfer setting and could potentially be applied elsewhere.

Here are some questions and comments related to this contribution:
- What is the motivation for the projection layer in the SSL loss? The choice is not ablated in the paper. I understand the need for such a layer in methods that do direct embedding supervision (either contrastive loss or simple L2 loss for example), but in this case the loss directly operates on the output distribution. Similar vision method like DINO [5] don't use such projector
- It would be valuable to provide "De Novo + Pisco" as a baseline to disentangle the gains from the transfer scheme from those due to the additional objective
- Though comparison to CURL is provided, SPR [6] has been shown to outperform it significantly, so a comparison seems warranted.




[1] "Big Transfer (BiT): General Visual Representation Learning", Kolesnikov et al, ECCV 2020

[2] "AdapterHub: A Framework for Adapting Transformers", Pfeiffer et al, EMNLP 2020

[3] "Learning multiple visual domains with residual adapters", Rebuffi et al, Neurips 2017

[4] "Universal Language Model Fine-tuning for Text Classification" Howard et al, ACL 2018

[5] "Emerging Properties in Self-Supervised Vision Transformers", Caron et al, ICCV 2021

[6] "Data-Efficient Reinforcement Learning with Self-Predictive Representations", Schwarzer et al, ICLR 2021



**Summary Of The Paper:**

This paper studies transfer learning in the context of Reinforcement Learning. Using 3 diverse domains as a testbed (a toy platform game, some tasks from the DeepMind Control Suite, and some navigation tasks from habitat), it makes the following contributions:
- Contribution #1: Shows that when transferring representations, the best strategy involves freezing the deepest layers and fine-tuning the others
- Contribution #2: Provides a strategy to select which layers to freeze and which to finetune
- Contribution #3: Introduces an auxiliary self-supervised objective to further enhance the transfer performance.




**Summary Of The Review:**

My main critiques for each of the contributions are as follows:

- The analysis of finetuning vs freezing representations misses the discussion on the capacity and pre-training diversity of the network involved, which in my opinion makes the picture incomplete
- The proposed strategy to choose which layers to freeze seems too costly, in its current form, to be broadly applicable
- The auxiliary objective is not studied in depth, be it with respect to its own design choices (eg the projection layer) or with respect to current state of the art auxiliary objectives (eg SPR)

As such, I don't recommend the paper in its current form for publication. That being said, all the aforementioned limitations seem fixable, and I am willing to increase my score if they are.

---

> ### Author Response · Authors · 2022-11-15
> **Response to Reviewer nmYq (part 1)**
>
> Thank you for bringing up an interesting interpretation of the results in Fig. 1. We’ve responded to that discussion, as well as the remaining of your concerns below.
>
> * The paper fails to make a broader conclusion given the results in Fig. 1: transfer succeeds (matches finetuning) when the network is large enough and pretrained on enough data.
>    * True, the Habitat representation extractor is much larger and trained on more data. However, we see Fig. 1.c-d highlighting a different failure mode: that finetuning fails to outperform naive representation freezing. In fact, the results in Fig. 5 support this observation since careful finetuning (with partial freezing / PiSCO) significantly outperforms frozen representations — despite the large architecture and large amount of pretraining data.
>    * So, *would transferring frozen representations succeed on Jump / DMC with more parameters and more data?* To test this hypothesis, we replicated the Frozen experiments using the same ConvNeXt as in the Habitat tasks. We report results in Appendix C.4 and observe that the larger network converges *slower* than the smaller networks.
>    * We explain those results as follows. The smaller networks are trained on data that is much more relevant to the downstream task, meaning they have a smaller generalization gap to bridge. In other words, more data is useful insofar as it is relevant to the downstream task.
>
>
> * Why use PCA on the intermediate features?
>    * To ensure our results are comparable across different layers. Early layers have many more features than latter ones, thus it’s significantly easier for linear models to fit arbitrary labels. Projecting all layers to the same number of features levels the field. Note that our analysis is robust to the number principal components (from 500 to 750, see footnote on p. 4).
>
>
> * Instead of action-values, could we predict the mean expected discounted return?
>    * Yes, most likely, given enough data. The advantage of learning a simple 2-layer action-value function is to remove some of the noise in the target signal; this reduces the amount of data required for our experiments.
>
>
> * Error bars are missing in Fig. 4.b (and similar figures in Appendix).
>    * Thanks for pointing this out; they are added and do not change the interpretation of the results (the confidence intervals are small).
>
>
> * What are the computational costs to identify layers-to-be-frozen? Do we need to finetune twice?
>    * No, we only need to finetune once; however, we do need to collect a little bit of extra data to perform the linear probing experiments. (Our experiments use 10k transitions, approx. 10-20 rollouts.)
>
>
> * Why does Fig. 11 not have estimates for finetuned representations?
>    * To illustrate that finetuned representations are not needed to identify which layers should be frozen: from that figure, it is clear that freezing up to Conv8 and Conv7 will yield best performance on Gibson and Matterport3D, respectively (due to the elbow shape of the graph). If Reviewer nmYq thinks it would help the reader, we are happy to include the Finetuned curves as well.
>
>
> * What is the motivation for the projection layer (no ablation)?
>    * The projector comes directly from the SimSiam formulation, and matters to get the best performance (both in their paper and ours). One intuition provided by the authors is that this projector helps prevent representation collapse. Due to your and Reviewer b3ER’s interest, we’ve added Figure 14 ablating the projector in Appendix C.6. The results clearly show that the projector is needed: without it, Frozen+PiSCO barely outperforms freezing all layers before transfer.
>
>
> * De Novo+PiSCO to disentangle the gains from freezing and PiSCO.
>    * Thank you for the suggestion. We’ve added a new figure 13 comparing De Novo+PiSCO to Frozen+PiSCO in Appendix C.5. On all DMC tasks, De Novo+PiSCO outperform De Novo alone, thus accelerating representation learning from scratch (as you initially hypothesized). We also observe that freezing task-agnostic layers plays an important part in obtaining the best out of transferred representations, as Frozen+PiSCO always outperforms De Novo+PiSCO. Both ingredients, frozen task-agnostic layers and self-supervised learning signal, are needed to get the best performance.

---

> > ### Author Response · Authors · 2022-11-15
> > **Response to Reviewer nmYq (part 2)**
> >
> >
> >
> >
> > * Add SPR as a baseline since it outperforms CURL.
> >    * Thank you for the suggestion. We’ve added new results in Appendix C.7 where we compare against Frozen+SPR in Figure 15. While Frozen+SPR underperforms Frozen+PiSCO (it approx. matches Frozen+Finetuned), we found an interesting result: we can improve upon Frozen+SPR by simply replacing the cosine similarity objective with the policy-induced objective suggested by our analysis (nicknamed PiSPR). Nonetheless, Frozen+PiSCO outperforms Frozen+PiSPR, is simpler to implement (fewer hyper-params, no transition model, no momentum weights), and is faster in wall-clock time (~3x). Still, we believe the SPR vs PiSPR results further validate our analysis insights and how they can generalize.
> >
> >
> > * What kind of data augmentation is used?
> >    * We use the data augmentation from the DrQ-v2 codebase, namely random shifts with 4px padding.

---

### Official Review · Reviewer_ZmQR · 2022-10-26

**Confidence:** 4
**Correctness:** 4
**Technical Novelty And Significance:** 2
**Empirical Novelty And Significance:** 2
**Recommendation:** 3

**Clarity, Quality, Novelty And Reproducibility:**

This paper is clearly written. However, suffers from a lack of novelty as I discuss above.

**Strength And Weaknesses:**

Strength:

- The paper is well-written and easy to follow.

Weaknesses:

- I am not an expert in reinforcement learning. However, the argument in the paper in terms of representation learning does not seem convincing and sufficient as an ICLR paper to me.  First, the author's argument about freezing vs finetuning in the introduction(namely the second and third paragraphs) holds for any neural network and downstream task. Posing it specifically as an RL problem does not seem rigorous to me.  Second, as it is mentioned by the authors themselves, the fact that the early layers of a convent capture general-purpose features while the later layers represent more task-specific features, is known and widely accepted. In fact, the pioneering work on fine-tuning such as Fast-RCNN achieves higher performance when the early layers are frozen, and the last layers are fine-tuned.

I can not evaluate the performance, but  I am not entirely convinced about the sufficiency of the contributions in. this paper.

**Summary Of The Paper:**

This paper studies fine-tuning vs freezing in the context of reinforcement learning. The authors argue that the layers that represent general-purpose features(mainly the early layers) could be frozen while those that represent task-specific features(mainly the last layers) should be fine-tuned to achieve higher performance in the downstream task.

**Summary Of The Review:**

In my opinion, this is a well-written paper that suffers from a lack of novelty and convincing arguments. Please see the above for more detailed feedback.

---

> ### Author Response · Authors · 2022-11-15
> **Response to Reviewer ZmQR**
>
> Thank you for your comments! We discuss your main concerns below, and justify why our contribution is relevant to ICLR and its RL community.
>
> * Why is freezing v.s. finetuning specific to RL?
>    * RL poses challenges that are not present in supervised or unsupervised finetuning; namely, the agent has to collect its own set of training data. The challenge stems from the policy having to explore the environment with random actions in the early stages of training, thus collecting noisy data which yield noisy gradients. Those noisy gradients (not present in SL or UL) can destroy the pretrained features during finetuning (Fig. 1.c-d); on the other hand, freezing the features does not work as well as in SL/UL (Fig. 1.a-b) for the reasons we uncover in Section 3: frozen features are more difficult to learn from (Fig. 3.b) and more fragile w.r.t. noise (Fig. 3.c).
>
> * It’s well-known that early layers are generic, latter ones specific.
>    * Yes, and our paper investigates: “how can we decide which features to freeze and which to finetune?”. We find that we can answer this question with a simple linear probing setup and a small amount of data from the downstream task.
>    * As for the claim in the question, we include a reference to Yosinski et al., 2014, which we believe is the first to establish this observation for deep nets. Please let us know if we missed an earlier reference. On the other hand, while the statement is generally true, investigating and clarifying  in RL, due to the difficulties mentioned above, is still valuable.
>
> * This paper suffers from a lack of novelty and convincing arguments for ICLR.
>    * We respectfully disagree with this assessment.
>    * First, note that other reviewers who are familiar with the RL literature have identified 3 novel contributions: 1) our analysis frozen and finetuned representations, 2) a strategy to identify which layers to freeze / finetune, and 3) a new method to stabilize RL finetuning. We hope these contributions can inform researchers, and inspire them to refine our studies and improve our methodologies.
>    * Second, we study a setting that is increasingly gaining traction in the RL community: transferring large models that are pretrained on generic datasets (see, e.g., Xiao et al, Parisi et al, Yamada et al, or Wang et al). In that setting, we show that *carefully* finetuning *some* layers can significantly improve performance over naive or no finetuning (see Fig. 5). This finding is particularly relevant to practitioners, who can benefit from significantly improved performance without paying the hefty cost of long RL training times.

---

### Author Response · Authors · 2022-11-15
**Summary of Reviews and Responses**

First, we would like to thank all reviewers for their comments on our submission. To help guide the discussion, we summarize some of the shared concerns below as well as the major updates we incorporated during the rebuttal period.


* **On motivation**: All reviewers expressed some concern over the setting of our study, namely taking pretrained representations and finetuning them with RL. We would like to emphasize the relevance of the setting, and the unique challenges it poses (some of which are uncovered in this submission).

    As Reviewer ZmQR points out, “freezing v.s. finetuning for transfer” is a horse beaten to death (especially in supervised learning for visual tasks); and yet, finetuning still doesn’t work in RL as demonstrated in Fig. 1. This is because in RL, unlike in SL, the agent has to collect its data which contains little learning signal (sparse/noisy rewards), especially in the early stages of finetuning. Consequence: the transferred knowledge is destroyed before the policy head was able to take advantage of it.

    Unfortunately, transfer is particularly crucial in RL due to sample inefficiency and the cost of collecting data. This trend is illustrated by the recent spark of interest in “RL as a finetuning paradigm” (eg, in RL from human preferences, pretrained vision models for control, SayCan, or embodied AI literature and codebases). So rather than proposing another pretraining objective, we assume the pretrained representations are given, study the finetuning problem, and reveal unintuitive situations where features that are good enough for SL but not for RL.
* **On empirical studies**: Following the reviewers’ suggestions, we have significantly expanded our analysis of our proposed PiSCO objective in the Appendix. The new results include:
   * Appendix C4: Does pretraining a feature extractor on generic but more data help with frozen representation transfer? Answer: No, it doesn’t because the generalization gap from generic to domain-specific data (as in DMC) hurts performance more than it helps.
   * Appendix C5: Are most of the gains in Frozen+PiSCO due to frozen layers or due to PiSCO? Answer: Both equally improve upon De Novo, and both are required for best performance.
   * Appendix C6: Do we need the projection layer in PiSCO (from SimSiam formulation)? Answer: Yes, removing it significantly degrades performance.
   * Appendix C7: How does PiSCO fare against SPR? Answer: Very well, while being easier to implement and faster to run. More interestingly: we can apply the insights from our analysis to improve SPR by replacing the cosine similarity objective with our proposed policy-induced objective. Those results validate the generality of our study.

---

### Decision · Program_Chairs · 2023-01-20

**Decision:**

Reject

**Justification For Why Not Higher Score:**

The paper needs major revision to make the story coherent and make the experiments clear and consistent.

**Justification For Why Not Lower Score:**

N/A

**Metareview: Summary, Strengths And Weaknesses:**

The goal of the paper is to study randomly initialized, fine-tuned and frozen backbones for a set of interactive tasks. Inspired by the study, a new self-supervised objective (Pisco) is proposed that provides better representations.

The paper is not written in a coherent way and it also suffers from lack of clarity and consistency. For example, (1) the paper mentions that "We regress the distance and optimal actions with linear models from evaluation task observations, using a mean square error and binary cross-entropy loss". It is not clear what the training data is. It seems it is trained on test data. It is not surprising that the task performance is 100% using either type of representation. (2) Some experiments are performed using a subset of the environments, some are done on all environments, and some are later added to the Appendix during the rebuttal period. These are just a few examples of lack consistency and clarity. There are various other issues similar to these.

Overall, the paper has many interesting analyses, and some of the approaches used for probing are interesting. However, it suffers from lack of coherency and clarity. There are several other weaknesses such as: (1) Some of the conclusions are presented as a general conclusion while the supporting experiments are done in a toy environment. (2) The computational cost of the approach is high (reviewer nmYq). (3) Some of the conclusions are known in the vision or RL communities (reviewer ZmQR). Despite the interesting analyses, the paper needs a major revision to address the coherency and clarity issues. Therefore, the AC recommends rejection.